# Nanoparticle-Mediated Therapeutic Application for Modulation of Lysosomal Ion Channels and Functions

**DOI:** 10.3390/pharmaceutics12030217

**Published:** 2020-03-02

**Authors:** Dongun Lee, Jeong Hee Hong

**Affiliations:** Department of Physiology, Lee Gil Ya Cancer and Diabetes Institute, College of Medicine, Gachon University, Incheon 21999, Korea; sppotato1@gmail.com

**Keywords:** nanoparticles, nanomaterials, lysosome, ion channels, nanodrugs

## Abstract

Applications of nanoparticles in various fields have been addressed. Nanomaterials serve as carriers for transporting conventional drugs or proteins through lysosomes to various cellular targets. The basic function of lysosomes is to trigger degradation of proteins and lipids. Understanding of lysosomal functions is essential for enhancing the efficacy of nanoparticles-mediated therapy and reducing the malfunctions of cellular metabolism. The lysosomal function is modulated by the movement of ions through various ion channels. Thus, in this review, we have focused on the recruited ion channels for lysosomal function, to understand the lysosomal modulation through the nanoparticles and its applications. In the future, lysosomal channels-based targets will expand the therapeutic application of nanoparticles-associated drugs.

## 1. Lysosomal Target of Nanoparticles (NPs) and Modulation of NPs for Lysosomal Function

### 1.1. pH Alteration

The primary function of the lysosome is the degradation of proteins and lipids [1,2]. The regulation of lysosomal pH has been linked to various cellular functions including the degradation of intracellular compartments. For its cellular functions, lysosomal lumen has to be maintained at an acidic pH [3]. Degradation of proteins, which is a crucial function of the lysosome, is carried out by more than 60 kinds of lysosomal hydrolases [4], and these hydrolases are optimized for the highly acidic environment of lysosomes (between pH 4.5 and 5.0) [4,5]. The lysosome as a cellular digestive system eliminates the garbage materials from autophagy and phagocytosis [6,7,8]. Thus, destabilization of lysosomal pH thorough alkalization leads to cellular toxicity and even causes lysosomal storage disease (LSD) [9,10,11]. The application of NPs can mediate various cellular functions by modulating lysosomal pH. Gold NPs (AuNPs) are known to reduce lysosomal activity by alkalization of the lysosomal lumen [11]. This reaction triggers oxidative stress, mitochondrial damage, and decreases cell migration/invasion [11]. In particular, 50-nm sized AuNPs induce autophagosomal accumulation of LC3 and block p62 degradation [12]. Silver NPs (AgNPs) also suppress autophagic responses by decreasing transcription factor EB (TFEB) protein expression, which is followed by lysosomal alkalization [13]. In addition, rare earth oxide NPs (REONPs)-mediated alkalization induces the activation of interleukin-1β IL-1β by an inflammasome [14].

### 1.2. Cell Viability

The lysosome consists of a typical single phospholipid bilayer to control important cellular functions [15,16]. The lysosomal membrane acts as the connector to contact other compartments such as autophagosome [17,18], mitochondria [19], and endoplasmic reticulum (ER) [20]. On the lysosomal membrane, numerous proteins play important roles such as the mammalian target of rapamycin complex 1 (mTORC1) (nutrient sensing) [21], V-ATPase (Vacuolar type of H^+^-ATPase) (pH homeostasis) [22], and ion channels/transporters [23]. In addition, deficiency of several lysosomal membrane proteins trigger various diseases such as the Danon disease (lysosome associated membrane proteins, LAMP-2) [24], malignant infantile osteopetrosis (the chloride channel 7, CLC-7) [25], and actin myoclonus-renal failure syndrome (lysosomal integral membrane protein-2) [26]. Damaged lysosome mediates lysosomal membrane permeabilization (LMP), which contributes to cell death [27,28] and induces several diseases such as LSD and other neurodegenerative disease [29,30,31]. Numerous NPs can have membrane damaging effects such as polystyrene NPs (PNPs) [32,33,34], silica dioxide (SiO_2_) NPs [35], titanium dioxide (TiO_2_) NPs [36], and Gd_2_O_3_:Eu^3+^ (Gd_2_O_3_) NPs [37], and, thus, cause cellular malfunctions. The PNPs (especially positive-charged) block autophagic flux [32], and release cathepsins (proteolytic enzymes), which induce cell death [34]. In addition, the LMP of other NPs reveal NACHT, LRR and PYD domains-containing protein 3 (NLRP3) inflammasome (SiO_2_NPs) [35] and necrosis (Gd_2_O_3_ NPs) [37].

### 1.3. Protein Activity and Expression

Various lysosomal functions are mediated by more than 200 integral lysosomal membrane proteins [4], including (1) the mechanistic target of mTORC1, which is activated by nutrient starvation [28,38], and acts as a negative regulator of autophagy [28,39], and (2) LAMPs, which protect the lysosomal membrane against lysosomal hydrolases not to degrade [40]. NPs induce an inhibitory effect on the mTORC1 pathway to activate autophagy: AgNPs (decreases lysosomal protease activities) [41], Zinc oxide (ZnO) NPs (induces macrophage cell death) [42], and REONPs (induces lysosomal imbalance by TFEB nucleus translocation) [43]. ZnO NPs induce an aberrant expression pattern and de-glycosylation of LAMP-2 by ZnO-induced reactive oxygen species (ROS), which trigger cell death in lung epithelial cells [44]. Additionally, NPs modulate lysosomal motility [45]. Lysosome movement reveals two directions: toward the peripheral cytoplasm (anterograde) [46,47] and juxtanuclear region (retrograde) [48]. To carry out autophagic flux, lysosomes have to move to the juxtanuclear region [22,38], and the dynein complex is the motor protein for retrograde transport [49]. Treatment with carbon nanotubes decreases the expression of synaptosomal-associated protein (SNAP), which is a regulating factor of dynein [50] that blocks retrograde transport and, thus, the autophagic pathway [45]. Taken together, the lysosomal pathways of NPs and occupied proteins may mediate numerous functions. Thus, careful and more extensive consideration of lysosomal-associated NPs needs to be done.

### 1.4. Accumulation of NPs

Toxic cellular components, such as cytoplasmic macromolecules, damaged or misfolded proteins, and other worn-out organelles, are removed by lysosomes to maintain metabolic homeostasis [3]. Thus, the degradation role of lysosomes is essential for carrying out cellular homeostasis [51] including lipid catabolism [52], cell growth [53], and neurotransmission [54]. However, several NPs interrupt lysosomal degradation and deposit the lysosomal compartment in the cytoplasm. Exposure to AgNPs and copper oxide (CuO) NPs can induce agglomeration of lysosomes and subsequent cellular damage, which leads to cell death in human lung alveolar epithelial cells [55] and human umbilical vein endothelial cells [56]. In addition, NPs can accumulate in lysosomes. SiO_2_NPs and PNPs impair cell viability and induce lysosomal swelling, which is followed by their accumulation in lysosomes and triggers lysosomal dysfunction and apoptosis [57,58].

## 2. Regulation of Lysosomal pH and Its Physiological Function

The lysosomal pH gradient is generated and maintained by movement of hydrogen ions (H^+^) into the lysosomes through the action of vacuolar-type ATPases (V-ATPases) [59], which is supplemented further by movement of other ions [5]. Thus, for effective and continuous movement of H^+^ into the lysosome, an accompanying counter-ion movement is necessary [5].

The lysosomal V-ATPases consists of two domains: V_1_ domain, which hydrolyses ATP, and the V_0_ domain, which translocates H^+^ ions across the lysosomal membrane [60]. The catalytic domain V_1,_ drives a rotary H^+^ transport motor by hydrolyzing ATP with translocation of H^+^ [61,62]. In this case, the V-ATPase rotor is operated in only one direction with an irreversible ATP hydrolysis due to the movement of H^+^ from cytosol to the lysosomal lumen [5]. The continuous V-ATPase-mediated H^+^ pumping generates a positive charge in the lysosomal lumen, which inhibits any further movement of H^+^ [63]. To dissipate this membrane potential, other ions have to be transferred in the opposite direction, and this process is referred to as the counterion flux [5,63]. Counter ion movement is suggested as both entering anions and exiting cations through the lysosomal lumen [5]. One important counter ionic candidate is chloride, transferred by CLC-7, as attenuation of CLC-7 leads to lysosomal dysfunction such as LSD and osteopetrosis [25,64]. Another candidate counter ion is K^+^, transferred by TMEM175. Its mutation induces neuronal degeneration and LSD [65]. The R740S mutant osteoclasts, mutated in the V-ATPase α3 subunit, possess a higher lysosomal pH, and shows altered mTORC expression (increase in basal protein level and decrease of gene expression) and activity, which, in turn, plays a key role in cell proliferation [57,66]. Additionally, acidification of lysosomes can induce macrophages to secrete *N*-acetyl-*β*-D-glucosaminidase through lysosomal exocytosis [67,68], which includes absorption of cytochrome c in rat kidney during renal metabolism [69], and transport of cystine, the product of protein degradation by cathepsin, from lysosomes to cytosol [70]. Thus, alteration of lysosomal pH can be like a commander’s order to modulate the cellular life cycle.

## 3. Lysosome-Associated Ion Channels for Lysosomal Function

The lysosomal function is modulated by the ion movement and subsequent pH regulation. This movement is accomplished through various ion channels (Figure 1). We have previously reported application of NPs on various channels [71]. In this section, we summarize the recruited channels for lysosomal function to understand the lysosomal modulation through the NPs (Table 1).

### 3.1. CLC

CLC channels are the chloride channels that play a critical role in lysosomal function. CLC channels consist of two major isotypes: plasma membrane-associated (CLC-1, -2, and -Ka/-Kb) and intracellular organelle-associated (CLC-3 to CLC-7) [114,115]. Among the intracellular organelle-associated CLC, CLC-3 channel promotes lysosomal acidification and induces bone resorption [72,73]. Deletion of CLC-6 DNA leads to LSD in neuronal cells [74]. Particularly, CLC-7 channel—a chloride/H^+^ antiporter—is a well characterized CLC channel that serves as a major pathway for chloride ion, and in lysosomes [116,117,118]. As mentioned above, CLC-7 has a regulatory role in lysosomes and inhibition of CLC-7 leads to various diseases such as LSD and osteopetrosis [25,64,81,82,83,84]. Lysosomal acidification is essential for osteoclast-mediated bone resorption. Mutations in the CLC-7 channel can inhibit the lysosomal acidification in an osteoclast [75,76] and trigger osteopetrosis [81,84]. CLC-7-deficient mice show LSD and neurodegeneration, which is followed by retinal degeneration [64,82]. For lysosomal acidification, the CLC-7 channel has to be trafficked to the lysosomes, supported by Ostm1 [119]. Acidification of lysosomes and activation of microglial cells both require CLC-7 channel trafficking to lysosomes for the degradation of amyloid-β peptide (fAβ) deposition, which drives Alzheimer’s disease (AD) [79,80]. Additionally, it has also been reported that deletion of the CLC-7 channel reduces the dentinogenesis and dental bone formation [77,78].

### 3.2. Cystic Fibrosis (CF) Transmembrane Conductance Regulator (CFTR)

CFTR is an ATP-binding protein, which is regulated by its phosphorylation regulatory (R) domains, and transports chloride among other anions including bicarbonate ion (HCO_3_^−^) [120,121,122,123,124]. Mutation of CFTR causes defects in fluid secretion and is responsible for the genetic disease CF [120,124,125]. A prevalent cause of CF results from a deletion of the 508th positioned phenylalanine (ΔF508) even though several other mutations have been identified in CF [120,121,125]. CFTR has been reported to support lysosomal acidification and is localized in intra-organellar components, including ER, Golgi, and endo/lysosomes [126,127]. In CF cells, which have a ΔF508 mutation in CFTR, lysosomal pH is higher than in normal cells [85]. CFTR-null macrophages showed a defective killing function of internalized bacteria by inhibiting phago-lysosomal fusion [86]. Typically, these macrophages kill bacteria by phago-lysosomal ingestion, which is followed by lysosomal acidification [86,127]. This suggests that CFTR-mediated lysosomal acidification can regulate bacteria-killing activity of macrophages. Additionally, activation of CFTR leads to re-acidification of alkalinized lysosomes in retinal pigmented epithelial cells, which suggests it is a useful target for lysosomal clearance [128].

### 3.3. TRPs

The TRP channels, grouped into six subfamilies of TRPC, TRPV, TRPM, TRPA, TRPP, and TRPML (transient receptor potentials canonical, vanilloid, melastatin, ankyrin, polycystic, and mucolipin, respectively), are cation permeable channels, composed of six transmembrane domains [129,130]. These channels, with their numerous subtypes, have various functions. In particular, TRPM2 and TRPML1-3 play important roles in lysosomes (only four subtypes are localized in the lysosomal membrane) [104,129,131].

#### 3.3.1. TRPM2

The TRPM2 channel is one of the TRPM family cation channels, which is activated by adenosine diphosphate ribose (ADPR) [132,133,134,135], adenine dinucleotide (NAD) [132,136], ROS [135,136,137], and extra/intra-cellular Ca^2+^ [138,139,140]. TRPM2 is located to numerous tissues and cellular compartments and has various activation mechanisms (Figure 2). Thus, the Ca^2+^ ion influx through TRPM2 plays multifunctional roles [141,142,143]. TRPM2 is also localized on the lysosomal membrane and modulates cellular functions such as cell migration, cytoskeleton remodeling, and apoptosis [87,88,89]. On the lysosomal membrane of dendritic cells (DC), TRPM2 releases Ca^2+^ ions to the cytoplasm to mediate optimal DC maturation and DC migration and homing to lymph nodes [87]. H_2_O_2_-induced Ca^2+^ influx increases through lysosomal TRPM2 and triggers actin remodeling, which, subsequently, activates cell migration, even though the extracellular Ca^2+^ entry does not affect the cytoskeletal remodeling [88]. Additionally, lysosomal TRPM2 Ca^2+^ ion release in pancreatic β cells induces apoptosis [89]. On the other hand, plasma membrane-localized TRPM2 mediates lysosomal damage via LMP and is associated with NLRP3 inflammasome-activation and mitochondrial fission [90,91].

#### 3.3.2. TRPMLs

TRPMLs (all three subtypes, TRPML1-3) are the main cation channels in the endo-lysosomal membrane, and regulate endo-lysosomal cation homeostasis, trafficking, and other cellular functions including intracellular compartment-acidification [104,131,144,145,146,147,148,149,150,151]. At the same time, TRPML1 is the main channel for lysosomal Ca^2+^ ion releases. TRPML2 and TRPML3 also have important roles in endosomal vesicles: regulation of TRPML2 is involved in the Arf6 recycling pathway [152], innate immune response [153], and B cell development [144,154]. The regulation of TRPML3 is involved in sensing lysosome neutralization [155], hearing functions [156,157], membrane trafficking, and autophagy [158]. Lysosomal Ca^2+^ ion-release through TRPML1 plays a major role in autophagy, mediated by starvation-induced mTORC1 deactivation and TFEB-induced autophagic gene expression [159,160] with simultaneous regulation of lysosomal acidification [92].

TRPML1 can regulate various cellular functions such as large particle phagocytosis through lysosomes [96], autophagosome biogenesis [161], elimination of bacterial pathogens through lysosome activation [162,163], bone remodeling in osteoclastogenesis [94], gastric acid secretion [93], and coronary arterial myocytes apoptosis [95]. In addition, the TRPML1 can reduce the enlargement of the lysosome by activating calmodulin [164]. Since TRPML1 has numerous functions, its deficiency can trigger various diseases, including stomach hypertrophy and hypergastrinemia [93], LSD [97,98,99], mucolipidosis type IV [100,101,102], Niemann-Pick disease type C (NPC) [97], and AD [103].

### 3.4. TMEM175

Intra-organelle K^+^ channel TMEM175 was recently identified in endosomes and lysosomes and is involved in the modulation of luminal pH stability and autophagosomes [165]. Deficiency of TMEM175 results in dysregulated lysosomal pH, impaired autophagosome clearance, and mitochondrial dysfunction in the neuronal system [166]. In addition to TMEM175, Ca^2+^-activated large conductance K^+^ channel also localizes to the lysosome and is involved in lysosomal Ca^2+^ signaling and lipid accumulation [104,105], which suggests lysosomal K^+^ channels can be considered the new target of neurodegenerative diseases such as LSD.

### 3.5. Other Ca^2+^ Channels

#### 3.5.1. Two Pore Channels (TPCs)

TPCs are the key components of Ca^2+^ signaling in the endo-lysosomal system including TRPML and TRP channels and have been extensively reviewed in various studies [167,168,169]. The TPC1-3 are identified in the endo-lysosome [170,171,172,173,174] and stimulated by nicotinic acid adenine dinucleotide phosphate and phosphatidylinositol 3,5-bisphosphate [171,175,176,177,178]. The roles and pathways of TPCs have been addressed in various organs and biological systems. The inhibition of the TPC channel abolishes the migration of metastatic cancer cells by disrupting the trafficking mechanism of β1-integrin and the formation of leading edges [179]. The TPCs are involved in the autophagic flux of mouse cardiomyocytes [180]. It has been discussed that TPC2 is involved in autophagy progression, cancer cell migration, and cellular pigmentation [106,107,108]. Additionally, signaling events of Parkinson’s disease involve the regulation of TPCs in trafficking [109,110].

#### 3.5.2. P2X4

The P2X4 receptor is expressed ubiquitously in cells from immune, nervous, muscle, and vascular systems [181,182,183]. The P2X4 is stable within the acidic environment of the lysosome and also traffics to the plasma membrane to enhance the phagocytic function [181,184]. P2X4 is activated by ATP and inhibited by the luminal acidic pH in the lysosome [185]. P2X4 consist of an ATP-activated Ca^2+^ channel and is involved in calmodulin activation to promote endo-lysosomal fusion of intracellular organelles [111,112]. P2X4 is also involved in liver fibrogenesis [113] and alcohol-induced microglial damage [186]. Although P2X4 has been associated with ATP-dependent signaling in the endo-lysosome, further studies are still needed in the future.

## 4. NP-Induced Proton Sponge Effect through Ion Channels in the Tumor System

Swelling of lysosomes has the potential to increase cellular toxicity by releasing lysosomal compartments and nanoparticles [187,188]. The lysosomal ‘proton sponge effect’ is triggered by the influx of cationic nanoparticles with hydrogen and chloride ions to lysosomes [188]. Accumulated ions in the lysosome may trigger water intake to equilibrate the physiological osmolarity and, subsequently, induce lysosomal rupture [188]. It has been addressed that conceptual use of the lysosomal pH-dependent system and lysosomal rupture develops the self-assembled luminescent AuNPs by the swelling property [189]. In a previous study, we reported that the cationic nanorod conjugated with doxorubicin (DOX) (AuNR-DOX) induced lysosomal swelling and rupture with increased apoptosis (Figure 3) [190]. Lee et al. reported that encapsulated AuNR-DOX in lysosomes is dissociated with DOX by lysosomal hydrolases. A charged linker of AuNR is opened and then recruited negative charged ions such as chloride into the lysosome. The ionic accumulation is developed, and lysosomal rupture occurred. Released chloride from the lysosome through lysosomal rupture activates Ca^2+^ influx channel TRPM2 in the plasma membrane and, lastly, overload of Ca^2+^ triggers the enhanced apoptotic effect including the effect of DOX in cancer cells [190]. The intracellular mechanism of nanomaterials and its related channels is now started. However, the effect of nanoparticles on lysosomal ion channels and transporters has still been poorly studied. To use nanomaterials for medicines, understanding the relationship between nanoparticles and lysosomal ion channels has to be expanded.

## 5. Clinical Application and Limitation of Nanomaterials

As mentioned earlier, NPs have a bio-toxic effect on lysosomes by triggering pH alteration, malfunctions of protein activity, accumulation in lysosomes, and subsequent cell death. We summarized the effect of NPs on cellular functions in Table 2. Accordingly, application of NPs has limitations for nanodrugs and nano-therapies. Thus, recent efforts have challenged to overcome these limitations by maximizing transport ability or reducing cytotoxicity.

Nanomaterials can act as the carrier for conventional drugs by transporting drugs or proteins through lysosomes such as AuNRs conjugated with Naja kaouthia protein toxin 1 (NKCT1) (one of the snake toxin protein) [191], silk NPs conjugated with doxorubicin (anti-cancer drugs) [192], and AgNPs conjugated with salinomycin (killing agent for cancer stem cells) [193]. These nanomaterials can maximize drug delivery to reach the lysosome easily, and, subsequently, kill the cancer cells from leukemia [191], breast cancer [192], and ovarian cancer [193]. The “small size” of NPs, which is one of the typical characteristics, can be used to penetrate obstacles that conventional drugs cannot cross, especially the blood brain barrier (BBB) [194]. One of the LSD, Gaucher’s type 3 disease, which occurs by accumulation of glucocerebroside in the brain can be cured by transporting enzymes into the brain [194]. A recent study demonstrated the potential for transporting enzymes across the BBB by using a recombinant arylsulfatase enzyme with polysorbate 80 coated poly-butyl cyanoacrylates NPs [195].

Although biocompatible nanodrugs have been developed, which are made of albumin-based [196,197,198] and lipid-based [199,200] nanoparticles, various studies have attempted to eliminate the toxicity of NPs via conjugation with other materials. For example, iron oxide NPs that induce autophagosome accumulation and impair lysosomes can be rendered bio-safe by coating with poly(lactic-*co*-glycolic acid) (PLGA) [193]. ZnO NPs and Quantum Dots that induce lysosomal damage with the generation of ROS can be stabilized by coating with α-linolenic acid [201] and 3-mercaptopropionic acid [202]. There are non-toxic nanomaterials that can be degraded into lysosomes, such as nano-diamonds, which are delivered to lysosomes by coating with ubiquitin, to associate with autophagy receptors: sequestosome 1 [203], Ca^2+^ binding and coiled-coil domain 2 [204,205], and optineurin [206]. Additionally, PLGA NPs are degraded easily in the autophagy pathway [207]. Adjustment of the NPs size can avoid lysosomal accumulation: 60 nm-sized TiO_2_ NPs are more aggregated and more destabilized in the lysosomal membrane than 180 nm-sized TiO_2_ in the lysosomes and endosomes [208].

## 6. Future Perspectives

The primary lysosomal function is to maintain cellular homeostasis. Various attempts of drug delivery systems including nanomaterials and other new paradigms against diseases were engaged (Figure 4). However, a plethora of questions should be answered about nano therapy against lysosomal targets or lysosomal pathways. Although our limited knowledge about the effect of nanomaterials on lysosomal function has been posted, its therapeutic potential cannot be neglected. Nanomaterials are attractive machinery, as carriers for conventional drugs for therapeutic purposes. In addition to the role of the attractive carrier, other unfavorable characteristics of nanomaterials including toxicity should be considered while developing the therapeutic strategies. Understanding the functional support of ion channels or transporters on the lysosome will be expanded further in the coming years and, subsequently, favorable potential of nanomaterial-based therapy will also improve.

## Figures and Tables

**Figure 1 pharmaceutics-12-00217-f001:**
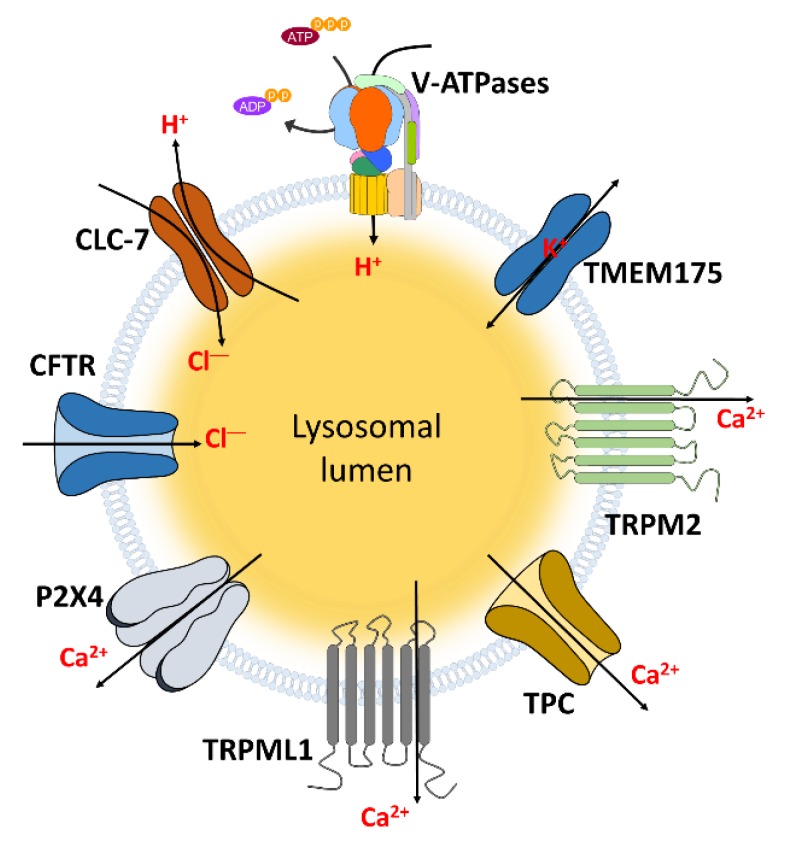
The channels localized in lysosomal membrane to transport ions. These channels and transporters can regulate lysosomal and cellular functions through transporting and maintaining hydrogen, chloride, Ca^2+^, and potassium which indicated in Table 1.

**Figure 2 pharmaceutics-12-00217-f002:**
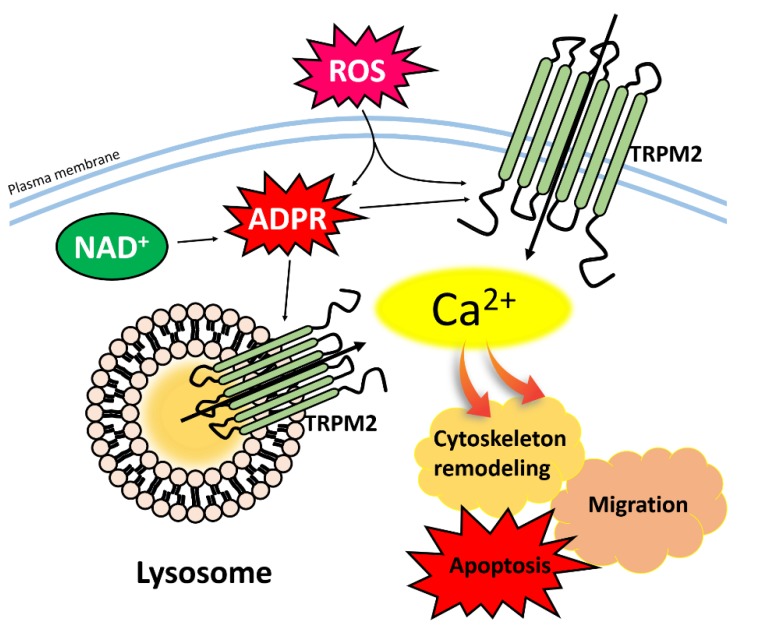
Activation of TRPM2 channel and cellular function. ADPR, NAD, and ROS induce up-regulation of intracellular Ca^2+^ concentration through the TRPM2 and, subsequently, mediate with cell migration, cytoskeleton remodeling, and apoptosis. Abbreviations: TRPM2: Transient receptor potentials melastatin 2; ADPR: Adenosine diphosphate ribose; NAD: Adenine dinucleotide; ROS: Reactive oxygen species.

**Figure 3 pharmaceutics-12-00217-f003:**
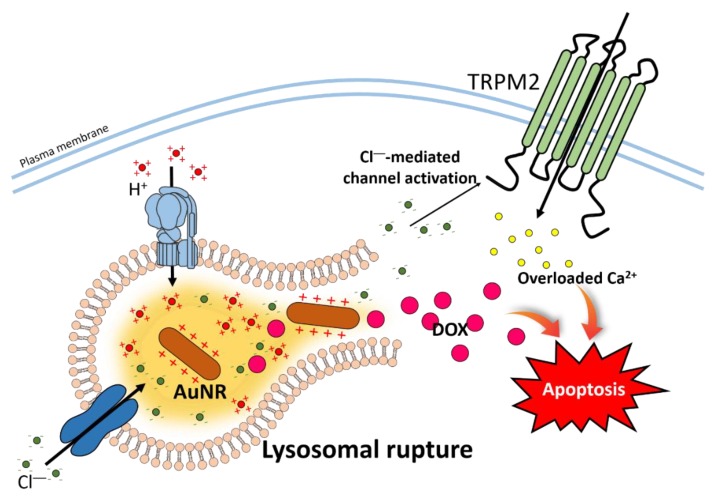
Schematic cartoon illustrating mechanism of AuNR-DOX-induced apoptosis. The hydrolysis of AuNR-DOX induces AuNR to reflect positive charge and triggers chloride influx into lysosomes. Continued chloride influx leads excessive activation of V-ATPase, and lysosomal swelling and rupture to release DOX and chloride to the cytoplasm. The DOX and Ca^2+^ through the chloride-activated TRPM2 increase cellular apoptosis. Abbreviation: V-ATPase: Vacuolar type of H^+^-ATPase.

**Figure 4 pharmaceutics-12-00217-f004:**
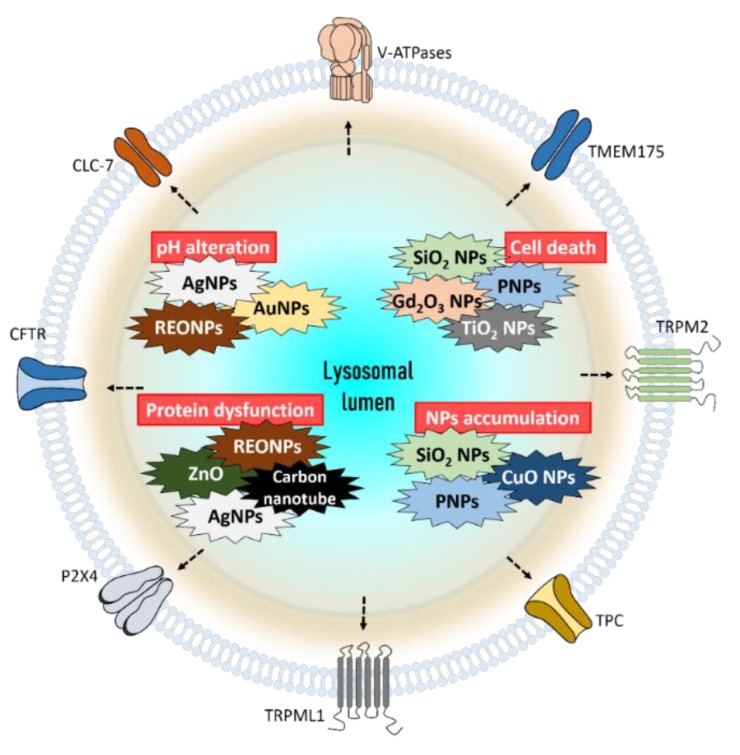
The summarized role of nanoparticles which effect lysosomes and cellular functions. Exposure to AgNPs, AuNPs, and REONPs induces alkalization of lysosomal lumen. SiO_2_ NPs, Gd_2_O_3_ NPs, PNPs, and TiO_2_ NPs damage lysosomes, which finally trigger cell death. The lysosomal protein can be damaged by ZnO, REONPs, AgNPs, and carbon nanotubes. Accumulation of SiO2 NPs, PNPs, and CuO NPs can induce lysosomal dysfunction. Abbreviations: REONP: rare earth oxide nanoparticle; PNP: polystyrene nanoparticle

**Table 1 pharmaceutics-12-00217-t001:** The relationship between lysosomal ion channels and cellular functions.

Channels	Mechanisms and Related Diseases	Ref.
CLC-3	Promotion of lysosomal acidification	[72,73]
CLC-6	LSD in CLC-6 mutated neuronal cells	[74]
CLC-7	Maintenance of acidic pH of lysosomes	[75,76]
	Decrease of dentinogenesis and dental bone formation in CLC-7 deficient mice	[77,78]
	Degradation of fAβ which drives AD	[79,80]
	Osteopetrosis in CLC-7 mutation	[81,82,83,84]
	LSD and neurodegeneration in CLC-7-deficient mice	[64,82]
CFTR	Support lysosomal acidification	[85]
	Decrease of bacteria killing function and phago-lysosomal fusion in macrophage	[86]
TRPM2	Induce DC maturation and migration	[87]
	Increase of actin remodeling	[88]
	Increase of pancreatic β cell apoptosis	[89]
	Increase LMP, NLRP3 inflammasome, and mitochondrial fission on the plasma membrane	[90,91]
TRPML1	Maintenance of acidic pH of lysosomes	[92]
	Increase of large particle phagocytosis, bone remodeling, gastric acid secretion, and myocytes apoptosis	[93,94,95,96]
	Stomach hypertrophy, hypergastrinemia, LSD, mucolipidosis, NPC, and AD in TRPML1 deficiency	[93,97,98,99,100,101,102,103]
TMEM175	Support lysosomal Ca2+ signaling and pH regulation	[104]
	Related in LSD	[105]
TPC	Related in autophagy, cancer cell migration, and cellular pigmentation	[106,107,108]
	Related in Parkinson’s disease	[109,110]
P2X4	Promotion of endo-lysosomal fusion	[111,112]
	Related in liver fibrogenesis	[113]

Abbreviations: CLC: Chloride channel; CFTR: Cystic fibrosis transmembrane conductance regulator; TRPM2: Transient receptor potential melastatin 2; TRPML1: Transient receptor potential mucolipin 1; TMEM175: Transmembrane protein 175; TPC: Two pore channel; AD: Alzheimer’s disease; DC: dendritic cell; LMP: Lysosomal membrane permeabilization; NLRP3: NACHT, LRR and PYD domains-containing protein 3; NPC: Niemann-Pick disease type C.

**Table 2 pharmaceutics-12-00217-t002:** The effect of nanoparticles (NPs) on cellular functions.

Related Cellular Function	NPs	Details	Reference
**pH alteration** **(alkalization of lysosome)**	AuNPs	Increase of oxidative stress, mitochondrial damage, and decrease cell migration/invasion	[11]
		Accumulation of LC3 and block p62 degradation	[12]
	AgNPs	Decrease of TFEB protein expression	[13]
	REONPs	Activation of IL-1β inflammasome	[14]
**Cell viability** **(cell death)**	PNPs	Decrease of autophagic flux	[32]
		Decrease of cathepsin release	[34]
	SiO_2_ NPs	Increase of membrane damage and NLRP inflammasome	[35,44]
	TiO_2_ NPs	Increase of membrane damage	[36]
	Gd_2_O_3_ NPs	Increase of membrane damage and necrosis	[37]
**Protein activity and expression**	AgNPs	Decrease of lysosomal protease activities	[41]
	REONPs	Induce lysosomal imbalance by inhibiting mTORC1 pathway	[43]
	ZnO NPs	Increase of macrophage cell death by inhibiting mTORC1 pathway	[42]
		Deglycosylation of LAMP-2	[44]
	Carbon nanotube	Decrease of SNAP	[50]
**Accumulation of NPs**	CuO NPs	Subsequent cellular damage leading to cell death by agglomeration of lysosomes	[55,56]
	SiO_2_ NPs, PNPs	Induce lysosomal swelling leading to apoptosis	[57,58]

Abbreviations: AuNP: Gold nanoparticle; AgNP: Silver nanoparticle; REONP: rare earth oxide nanoparticle; PNP: polystyrene nanoparticle; ZnO: Zinc oxide; CuO: Copper oxide; TFEB: Transcription factor EB; IL-1β: interleukin-1β; NLRP: NACHT, LRR and PYD domains-containing protein; mTORC1: rapamycin complex 1; SNAP: synaptosomal-associated protein.

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
