# Peer review of "Nanoparticle-Mediated Therapeutic Application for Modulation of Lysosomal Ion Channels and Functions"

_pharmaceutics, 2020, doi:10.3390/pharmaceutics12030217_

Round 1
Reviewer 1 Report
Comments
The authors Lee et al provided an overview of the lysosomal functions and their therapeutic application through the use of ion channels lysosomal-targeted nanoparticles. In particular the authors focus on the lysosomes as a target of nanoparticles and their modulation in activity. Moreover, their described the important role of ion channels for lysosomal function. Finally, they look at the role of nanoparticles in inducing proton sponge effect via ion channels in tumor. They conclude with nanoparticles clinical application, limitation and future perspective.
This is a well documented work and has a good relevant to the field.
The manuscript would benefit from the following:
In the section 1.3 "Protein activity and expression", the authors should discuss about the role of LAPTM4a and LAPTM4B both involved in the transport of small molecules across endosomal and lysosomal membranes. Previous works have linked their role to chemotherapy resistance, an example is the paper titled Primary Culture of Undifferentiated Pleomorphic Sarcoma: Molecular Characterization and Response to Anticancer Agents. Int J Mol Sci. 2017 Dec 8;18(12). pii: E2662. doi: 10.3390/ijms18122662, which should be included in this section. In the section 5. "Clinical application and limitation of nanomaterials", no metion is made of albumin-based and lipid-based nanoparticles which are already used patient care management. The authors should include in the reference the manuscript titled: Lysosomal proteins in cell death and autophagy. FEBS J. 2015 May;282(10):1858-70. doi: 10.1111/febs.13253. Epub 2015 Mar 23. which represent an overview of the role of lysosomes.
Minor corrections are required before publication
Author Response
Dear reviewers and chief editor,
Before addressing each of the comments below, we appreciate the reviewers for the valuable comments and careful consideration. The manuscript has been edited to make appropriate information to this body of work.
Responses to comments of reviewers as below:
Reviewer#1
The authors Lee et al provided an overview of the lysosomal functions and their therapeutic application through the use of ion channels lysosomal-targeted nanoparticles. In particular, the authors focus on the lysosomes as a target of nanoparticles and their modulation in activity. Moreover, their described the important role of ion channels for lysosomal function. Finally, they look at the role of nanoparticles in inducing proton sponge effect via ion channels in tumor. They conclude with nanoparticles clinical application, limitation and future perspective.
This is a well documented work and has a good relevant to the field.
- Response: We appreciate your valuable comments.
The manuscript would benefit from the following:
In the section 1.3 "Protein activity and expression", the authors should discuss about the role of LAPTM4a and LAPTM4B both involved in the transport of small molecules across endosomal and lysosomal membranes. Previous works have linked their role to chemotherapy resistance, an example is the paper titled Primary Culture of Undifferentiated Pleomorphic Sarcoma: Molecular Characterization and Response to Anticancer Agents. Int J Mol Sci. 2017 Dec 8;18(12). pii: E2662. doi: 10.3390/ijms18122662, which should be included in this section.
- Response: We appreciate your comment. In the section 1.3, we tried to summarize the relationship between lysosomal proteins and nanoparticles. We recognized that the LAPTM4a and the LAPTM4B are important proteins for lysosomes, but we found that they are not connected to nanoparticles. In this review, we focused on the relationship between lysosomes and nanoparticles through only the transports of electrolytes and ions not small molecules. We appreciate to recommend new proteins, but we hope to concentrate our subject.
In the section 5. "Clinical application and limitation of nanomaterials", no metion is made of albumin-based and lipid-based nanoparticles which are already used patient care management.
- Response: We appreciate your comments. The manuscript has been edited as you recommended in page 5.
The authors should include in the reference the manuscript titled: Lysosomal proteins in cell death and autophagy. FEBS J. 2015 May;282(10):1858-70. doi: 10.1111/febs.13253. Epub 2015 Mar 23. which represent an overview of the role of lysosomes.
- Response: We appreciate your comments. The reference has been added as you recommended in page 2.
Reviewer 2 Report
Pros:
- Authors have covered all aspects of Lysosomal ion channels
- They have given extensive reference from all published works on Ion channel
- Nanoparticle approach in Lysosomal modulation is well covered
Cons:
- One of the major functions of Lysosomes is the acidification of phagosomes and the elimination of pathogens. The authors have completely ignored this part.
- Nanoparticles such as AgNPs, AuNPs are successfully used in controlling biofilms of some superbugs such as Acinetobacter and Pseudomonas. Authors have failed to give any importance to such areas
- Authors have not included some of the recent works which have connected TRPMLs with TFEB in with and without infection. For example Rosato et.al 2019, Nat.Com, Capurro MI et al 2019, Nat. Mic, Yuxuan Miao et al 2015, Cell, Xiaopeng Qi et al 2016, JEM.
- It will be good if the authors can address some of these findings in the context of nanoparticle therapy
Author Response
Dear reviewers and chief editor,
Before addressing each of the comments below, we appreciate the reviewers for the valuable comments and careful consideration. The manuscript has been edited to make appropriate information to this body of work.
Responses to comments of reviewers as below:
Reviewer#2
Minor corrections are required before publication
Pros:
Authors have covered all aspects of Lysosomal ion channels
They have given extensive reference from all published works on Ion channel
Nanoparticle approach in Lysosomal modulation is well covered
- Response: We appreciate your valuable comments.
Cons:
One of the major functions of Lysosomes is the acidification of phagosomes and the elimination of pathogens. The authors have completely ignored this part.
- Response: We appreciate your comments. We added the major function of lysosome and the manuscript has been edited as you recommended in page 1.
Nanoparticles such as AgNPs, AuNPs are successfully used in controlling biofilms of some superbugs such as Acinetobacter and Pseudomonas. Authors have failed to give any importance to such areas
Authors have not included some of the recent works which have connected TRPMLs with TFEB in with and without infection. For example Rosato et.al 2019, Nat.Com, Capurro MI et al 2019, Nat. Mic, Yuxuan Miao et al 2015, Cell, Xiaopeng Qi et al 2016, JEM.
It will be good if the authors can address some of these findings in the context of nanoparticle therapy.
- Response: We appreciate your valuable comment. However, in this review, we focused on the relationship between lysosomes and nanoparticles through only the transports of electrolytes and ions. The anti-bacterial effect of AgNPs and AuNPs is interesting and phagosome and its related field are pretty attractive, but it is too extensive to include the anti-bacterial effect of nanoparticles. It is a good idea to write another review about nanoparticle and anti-bacterial subject. We include the recommended references in TRPML section and brief explanation. We again appreciate your recommendation, but we hope to concentrate our main concept.